# A Novel Silicon Forward-Biased PIN Mach–Zehnder Modulator with Two Operating States

**DOI:** 10.3390/mi14081608

**Published:** 2023-08-15

**Authors:** Hang Yu, Donghe Tu, Xingrui Huang, Yuxiang Yin, Zhiguo Yu, Huan Guan, Lei Jiang, Zhiyong Li

**Affiliations:** 1State Key Laboratory on Integrated Optoelectronics, Institute of Semiconductors, Chinese Academy of Sciences, Beijing 100083, China; 2College of Materials Science and Opto-Electronic Technology, University of Chinese Academy of Sciences, Beijing 100083, China

**Keywords:** silicon photonics, optical modulator, two operating states, passive equalizer

## Abstract

In this paper, we demonstrate a silicon forward-biased positive intrinsic negative (PIN) Mach–Zehnder modulator (MZM), which has two operating states of high efficiency and high speed. The two operating states are switched by changing the position where the electric signal is loaded. The modulator incorporates a PIN phase shifter integrated with the passive resistance and capacitance (RC) equalizer (PIN-RC), which expands the electro-optic (E-O) bandwidth by equalizing it with modulation efficiency. The fabricated modulator exhibits a low insertion loss of 1.29 dB in two operating states and a compact design with a phase shifter length of 500 μm. The modulation efficiencies are 0.0088 V·cm and 1.43 V·cm, and the corresponding 3 dB E-O bandwidths are 200 MHz and 7 GHz, respectively. The high-speed modulation performance of the modulator is confirmed by non-return-to-zero (NRZ) modulation with a data rate of 15 Gbps without any pre-emphasis or post-processing. The presented modulator shows functional flexibility, low insertion loss, and a compact footprint, and it can be suitable for applications like optical switch arrays and analog signal processing.

## 1. Introduction

In recent years, silicon photonics has emerged as a proposed technology to achieve ultra-compact integrated optical devices on a single chip because of its potential advantages, such as the mature complementary metal oxide semiconductor (CMOS) process and strong optical confinement in the SOI waveguide [1,2]. High-performance modulators are critical components in photonic integrated circuits. In particular, MZMs are widely used due to their excellent characteristics of high modulation, broadband operational spectra, and high thermal stability, which are being utilized as a fundamental building block in several photonic integrated circuits [3,4,5,6,7].

Over the past decades, many kinds of modulators based on silicon have been demonstrated. Among them, the most common method is to exploit the plasma dispersion effect of silicon [8,9,10,11,12,13,14,15] since it does not need to integrate other materials like lithium niobate (LN) [16,17,18,19] or electro-optical polymer [20,21,22]. The refractive index of silicon varies with the carrier concentration inside the material, which is mainly achieved through two methods: carrier depletion (reversed-biased) [23,24,25,26,27] and carrier injection (forward-biased) [28,29,30,31]. To date, in the field of high-speed transmission, the research on carrier-depletion-based MZMs is far more than carrier-injection-based MZMs for high-speed transmission. The main reason is that carrier-depletion-based MZMs are suitable for high-speed applications when velocity matching and impedance matching are achieved [32,33,34]. However, the active length of carrier-depletion-based MZMs is 1 mm or longer, which increases the difficulty of large-scale integration [35]. Although modulation efficiency can be improved by using a high capacitance with heavy doping of PN junction [36], it will introduce extra insertion loss. PIN modulators have a higher modulation efficiency and smaller size under forward bias [37]. However, the difficulty in the practical application of PIN modulators is a characteristic of frequency response, and the 3 dB electro-optical bandwidth is usually hundreds of MHz due to large diffusion capacitance [38]. The most common method to increase the bandwidth is to load pre-emphasis electrical signals to compensate for the attenuation of the modulator’s high-frequency response. In ref. [39], pre-emphasis signals were successfully introduced to compensate for the frequency dependence, and improve the modulation rate up to 25 Gbps. In [40], a silicon forward-biased PIN microring modulator is demonstrated, with a data rate of up to 56 Gbps by loading pre-emphasis signals. Although using pre-emphasis signals is an effective method to improve the high frequency of the PIN modulator, it needs an extra digital signal processor (DSP) to generate pre-emphasis signals, which increases the difficulty of driving circuits. In [41], the E-O bandwidth of the PIN modulator is greatly improved by integrating a passive RC equalizer, which reduced the equivalent capacitance of the circuit. This method does not need to modify the shape of the electronic signal, and no additional power consumption occurs. Hence, forward-bias-based PIN modulators not only confer distinct advantages in terms of modulation efficiency, insertion loss, and footprint, but they also manifest significant potential in the realm of modulation speed.

In this paper, we design, model, and characterize the silicon forward-biased PIN modulator integrated with passive RC equalizers. The PIN-RC modulator has two operating states, which are the high-efficiency state and the high-speed state. The two operating states are switched by the position where the electric signal is loaded. The demonstrated modulator exhibits a low insertion loss of 1.29 dB in two operating states and a compact design with a phase shifter length of 500 μm. The modulation efficiencies in the two operation states are 0.0088 V·cm and 1.43 V·cm, and the 3 dB E-O bandwidths are 200 MHz and 7 GHz, respectively. In the high-speed operation state, the eye diagrams of 10 Gbps and 15 Gbps NRZ signals are demonstrated without any pre-emphasis or post-processing. Moreover, the SOA damages the signal-to-noiseratio (SNR), which also has a negative effect on the quality of the eye diagram.

## 2. Structure and Design

A schematic view of the proposed PIN-RC modulator is shown in Figure 1. Two grating couplers (GCs) designed for the transverse electric (TE) polarization at 1310 nm are used for off-chip coupling. The PIN-RC modulator is designed to be asymmetrical; thus, effective orthogonality of the outputs from the phase shifters can be realized at a proper wavelength. An equal power splitter is achieved based on a 1×2 multimode interferometers (MMIs). The modulator has a compact design with a phase shifter length of 500 μm. The passive RC equalizer is cascaded with the PIN phase shifter to expand the E-O bandwidth of the modulator. The metal pads at both ends of the modulator are used to load the electric signal.

The demonstrated PIN-RC modulator has two modes of operation, which are switched by the position where the electric signal is loaded. When the electric signal is loaded to the PIN-RC modulator from metal pad1, the modulator shows high-efficiency characteristics. The modulator shows high-speed characteristics when the electric signal is loaded from the metal pad2.

The principle of the PIN-RC modulator can be explained by the equivalent circuit model of the modulator, as shown in Figure 2. The modulator has a short electrode relative to the RF wavelength and can be regarded as a lumped electrode, which considers the velocity mismatch and the impendence mismatch little. R${}_{drv}$ is the impedance of the driver, R${}_{s}$ is the series resistance, and C${}_{F}$ and R${}_{F}$ are the equivalent capacitance and resistance of the PIN diode, respectively. C${}_{BOX}$ is the capacitance of the buried S${}_{i}$O${}_{2}$ (BOX) layer, and C${}_{sub}$ and R${}_{sub}$ are the capacitance between the metal electrode and the silicon substrate and the transverse resistance of the silicon substrate, respectively. Figure 2a shows the equivalent circuit of the PIN-RC modulator when the RF signal is loaded to metal pad1. The RC equalizer is open, and the un-equalized modulator resembles a conventional PIN modulator with high modulation efficiency and low E-O bandwidth. When the electric signal is loaded to the modulator from metal pad2, the equivalent circuit is shown in Figure 2b. The RC equalizer can work normally, and the E-O bandwidth of the equalized modulator is improved. The essence of the RC equalizer is a high-pass filter, which compensates for the attenuation of the modulator output signal at high frequency. The RC equalizer comprises a parallel-arranged capacitance, C${}_{E}$, and resistance, R${}_{E}$, and they are satisfied with the following equation:(1)$$C_{E}=\frac{C_{F}}{\eta },\;R_{E}=R_{F}\times \eta ,\;\eta \gg 1$$

According to the equation, the capacitance, C${}_{E}$, is much smaller than C${}_{F}$, and the equivalent capacitance of the circuit is greatly reduced because of the cascading of capacitors. Therefore, the E-O bandwidth of the PIN modulator is improved. In addition, we set the matching condition of the equalizer, R${}_{E}$C${}_{E}$ = R${}_{F}$C${}_{F}$, to obtain a flat S${}_{21}$ frequency response. There is a sharp peak or valley when the matching condition is not satisfied, and the presence of sharp peaks or valleys in the frequency response can lead to an overshoot in low-frequency regions, thereby compromising the integrity of modulated signals.

In this work, the length of the phase shifter is 500 μm, and we calculated the C-V characteristic and I-V characteristic of the forward-biased PIN junction to obtain the electrical parameters, as shown in Figure 3. We chose the vicinity of the position with the maximum slope of the C-V curve as the direct-current (DC) bias point (0.84 V), and the corresponding diffusion capacitance (C${}_{F}$) is 28.05 pF. According to the I-V characteristic of the PIN junction, the sum of the diffusion resistance (R${}_{F}$) and the parasitic resistance (R${}_{s}$) is 55.2 Ω, and we calculated the parasitic resistance (R${}_{s}$) to be 5.2 Ω. The factor of η is set to be 340, and the capacitance, C${}_{E}$, and resistance, R${}_{E}$, of the RC equalizer are 82.5 fF and 17 kΩ, respectively.

Figure 4 shows the simulated small-signal E-O responses of the PIN-RC modulator in two operation states, respectively. When the PIN-RC modulator is operating in the high-speed state, the E-O response exhibits a flat response, and the 3 dB E-O bandwidth is up to 7 GHz. The bandwidth of the PIN-RC modulator in the high-efficiency state is 200 MHz. In addition, the equalization technique requires no extra power because the RC equalizer is passive.

It is worth noting that the parasitic parameters of the substrate seriously degrade the E-O bandwidth. As shown in Figure 2, the capacitors C${}_{BOX}$ and C${}_{sub}$ increase the total capacitance of the circuit, and thus, the effective multiplication factor for the bandwidth is reduced. Figure 4 also shows the simulated frequency responses of the equalized PIN-RC modulator without substrate parasitic parameters, and the calculated E-O S21 shows about −1.19 dB drop from DC to 20 GHz, and the simulated 3 dB E-O bandwidth is 35 GHz. Therefore, we believe that the bandwidth of the equalized PIN-RC modulator is particularly sensitive to the parasitic parameters of the substrate.

Figure 5 shows the cross-sectional structure of the equalized PIN phase shifter. The resistance, R${}_{E}$, is realized by the doped silicon waveguide, and we can extract the resistivity and calculate the size according to the formula. The capacitance, C${}_{E}$, is realized by the metal insulator metal (MIM) structure. The thicknesses of the top and bottom metal layers are 850 nm and 450 nm, and the gap between the two metal layers is 450 nm, which is filled with S${}_{i}$O${}_{2}$. We used electromagnetic simulated software to calculate the S parameters and fit the size of the capacitor. The passive RC equalizer has a very compact footprint of 26 × 30 μm${}^{2}$, and it can be placed close to the PIN phase shifter.

While ensuring the high-speed operation of the modulator, it is also important to consider the optical loss characteristics. The optical loss of the silicon modulator is mainly caused by the free carrier absorption effect. For the forward-biased PIN-RC modulator, the optical loss is mainly determined by the dopant distributions and the applied voltage. Figure 5 shows the cross-sectional structure of the PIN phase shifter, the silicon waveguide is 400 nm wide and 220 nm high with an etched depth of 150 nm, and the width of the intrinsic layer is 360 nm. The P++ and N++ doped regions on both sides are designed to reduce the ohmic contact resistance with the metal electrodes, and the two regions are 380 nm away from the boundary of the ridge waveguide, which almost eliminates the absorption of the light field by the heavily doped regions. The N+ and P+ doped regions are doped with 1.15 × $10^{18}$ cm${}^{-3}$ and 8.5 × $10^{17}$ cm${}^{-3}$, respectively. Two narrow (20 nm) regions at both sidewalls of the ridge waveguide are intentionally doped to transport carriers to the upper side of the intrinsic region. In addition, the optical loss of the PIN-RC modulator increases with applied voltage due to the forward injection of carriers. Therefore, the DC bias point of the modulator is selected near the forward conduction voltage of the PIN diode, which can ensure high modulation efficiency and low optical loss.

## 3. Experiments and Results

The demonstrated device was designed and fabricated on the SOI platform with a 220 nm thick top-silicon and a 2 μm thick BOX layer. Figure 6 shows the microscope image of the PIN-RC modulator and its close-up. The modulator operates in O-band. The metal pads are distributed at both ends of the modulator to load the electric signal, and the two operating states are switched by the position where the electric signal is loaded.

The DC characterization was performed first. A tunable laser was used as the source. A polarization controller (PC) was added to control the polarization state of the input optical signals. The measured coupling loss of the grating coupler was 3 dB/facet. The insertion loss of the PIN-RC modulator was extracted by using the cut-back method. The total on-chip insertion loss of the PIN-RC modulator in two operating states was about 1.29 dB. It is noticed that the insertion loss is measured in the ON state. For the carrier-injection-based MZMs, the insertion loss increases with the voltage due to the forward injection of carriers. Therefore, the modulator has the lowest insertion loss in the OFF state, and the measured insertion loss is about 0.64 dB near 1310 nm. In fact, the insertion losses of the two states are not exactly the same, and there is a slight difference between the two states. Accounting for the impact of measurement errors and the uniformity of the grating couplers, we make an approximation that the overall loss experienced by both states is approximately 1.29 dB. The main reason for the close loss of the two states is that the DC bias points in the two states are very close, and thus, the difference in the amount of carrier injection is small. Moreover, it is imperative to underscore that the extent of the phase shifter spans a mere distance of 500 μm, thereby mitigating the substantial repercussions posed by fluctuations in loss incidence per unit length upon the overarching cumulative loss profile. Afterward, we applied low-frequency triangle wave signals with different drive voltages to the modulator and then used an oscilloscope to measure the output optical power, and the measured V${}_{\pi }$ L${}_{\pi }$ in two operation states were 0.0088 V·cm and 1.43 V·cm, respectively.

Then, characterizations of small signals were performed. Figure 7a shows a schematic of the experimental setup for the small-signal characterization of the PIN-RC modulator. A tunable laser fixed at 1310 nm was used as the optical source. The output optical signal was detected by a high-bandwidth E-O bandwidth photodetector (PD), which was connected to the vector network analyzer (VNA). A 67 GHz GS high-frequency RF probe was used to launch the RF signal. By de-embedding the know E-O S21 parameters of the PD, the E-O response of the proposed PIN-RC modulator was obtained. The measured E-O S21 is shown in Figure 7b. The 3 dB E-O bandwidths of the PIN-RC modulator in two operating states are 200 MHz and 7 GHz. It can be seen that the modulator has a flat E-O response in the high-speed state. The experimental results were similar to the simulated values. The RC-equalized technique to expand the analog bandwidth was successfully verified.

The influence of the parasitic parameters of the substrate is one of the main factors limiting an improvement in the E-O bandwidth. The demonstrated modulator was fabricated on a low-resistivity silicon substrate with a resistivity of about 16.5 Ω·cm. According to the equivalent circuit model of the PIN-RC modulator, the influence of the substrate capacitance, C${}_{sub}$, can be reduced by increasing the resistivity of the substrate; thus, the E-O bandwidth of the PIN-RC modulator can be further increased.

Next, we studied the large-signal performance of the PIN-RC modulator in the high-speed operating state. Eye diagrams of NRZ signals with different data rates were measured. A pseudo-random binary sequence (PRBS) NRZ signal of $2^{31}$-1 was generated by a bit pattern generator and amplified by a microwave amplifier. The input optical signal was amplified with a semiconductor optical amplifier (SOA), and the output optical fiber was directly connected to a digital serial analyzer (Tektronix DSA8300). The input wavelength was set to the quadrature point while using the asymmetric MZI. Figure 8a,b show the measured eye diagrams with the data rate of 3 Gbps and 5 Gbps, respectively. The peak-to-peak amplitudes ($V_{pp}$) of the drive signals were measured to be 7.83 V and 7.92 V, and the corresponding extinction ratios (ERs) were 3.27 dB and 3.18 dB, respectively. Due to the bandwidth limitation and nonlinearity of the RF amplifier, the values of $V_{pp}$ decreased with the increase in the data rate, and the ER of the eye diagram also decreased. The eye diagrams with the data rates of 10 Gbps and 15 Gbps are shown in Figure 8c,d. The $V_{pp}$ of the drive signal of both cases was 4.01 V, and the corresponding ERs were 1.54 dB and 1.06 dB. Note that all the eye diagrams were obtained without any digital signal processing or post-compensating. The SOA damages the signal-to-noise ratio (SNR), which also has a negative effect on the quality of the eye diagram. Moreover, the quality of the eye diagrams with data rates of 10 Gbps and 15 Gbps can be further increased by utilizing a driver with large swings.

## 4. Discussion and Conclusions

The demonstrated PIN-RC modulator shows functional flexibility. When the modulator is operating in the high-efficiency state, the device exhibits a high modulation efficiency of 0.0088 V·cm. On the other hand, the modulator enables 15 Gbps of high-speed data transmission when the modulator is operating in the high-speed operating state. Table 1 summarizes several reported works on silicon forward-bias-based PIN Mach–Zehnder modulators. It can be seen that the performance of the modulator is not the best, especially in terms of modulation speed. Compared with ref [41], we conclude that the main reason for the significant reduction in the E-O bandwidth is that the modulator was fabricated on a low-resistance silicon substrate, and the large parasitic parameters degraded the bandwidth of the device. Analyzing the influence of parasitic parameters on the modulation speed will be the focus of our next research. Overall, the simultaneous realization of the two operating states extends the application range of the modulator, and the characterizations of low loss and compact footprint exhibit the potential for large-scale integration.

In summary, we designed, modeled, and characterized a silicon forward-biased PIN modulator integrated with passive RC equalizers. The demonstrated PIN-RC modulator has two operation states of high efficiency and high speed, which are switched by changing the position where the electric signal is loaded. The proposed modulator has a low insertion loss of 1.29 dB in two operating states and a compact design with a phase shifter length of 500 μm. The measured modulation efficiencies in the two operation states are 0.0088 V·cm and 1.43 V·cm, and the corresponding 3 dB E-O bandwidths are 200 MHz and 7 GHz, respectively. High-speed data transmissions of 10 Gbps and 15 Gbps can also be achieved. The demonstrated modulator can be switched flexibly between two operating states by changing the position where the electric signal is loaded. In conclusion, the demonstrated modulator shows its functional flexibility, low insertion loss, and compact footprint characteristics, and it has the potential to be applied to optical switch arrays, analog signal processing, and optical communication links.

## Figures and Tables

**Figure 1 micromachines-14-01608-f001:**
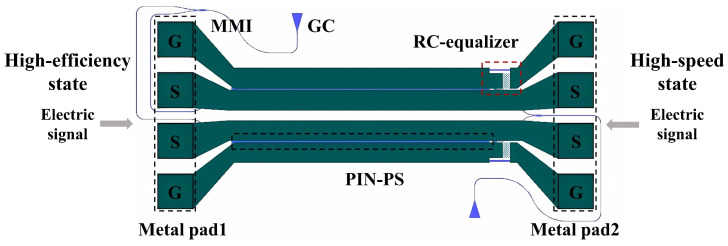
Schematic view of the proposed PIN-RC modulator integrated with RC equalizer.

**Figure 2 micromachines-14-01608-f002:**
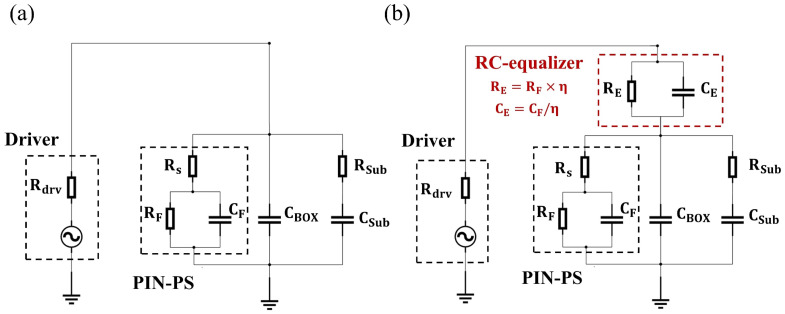
Equivalent circuit of the forward-biased PIN-RC modulator in the high-efficiency state (**a**) and high-speed state (**b**).

**Figure 3 micromachines-14-01608-f003:**
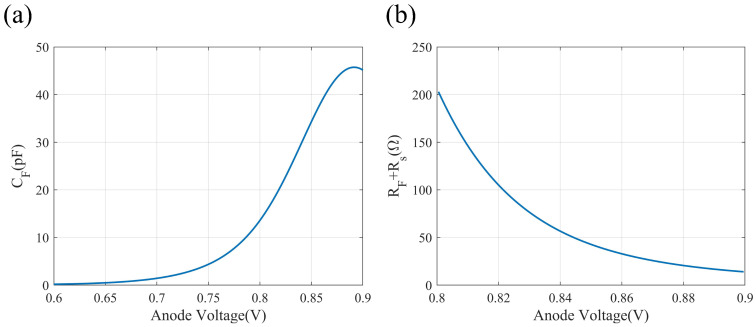
C-V characteristic (**a**) and I-V characteristic (**b**) curves of the forward-bias PIN junction.

**Figure 4 micromachines-14-01608-f004:**
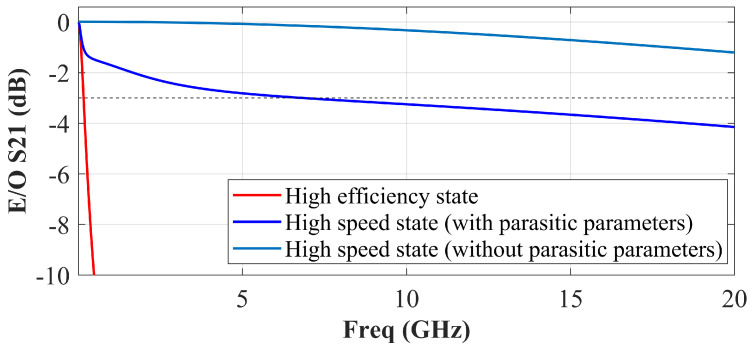
Simulated E-O responses of the PIN-RC modulator in the high-efficiency state (red line), high-speed state (with parasitic parameters; blue line), and high-speed state (without parasitic parameters; pale blue line).

**Figure 5 micromachines-14-01608-f005:**
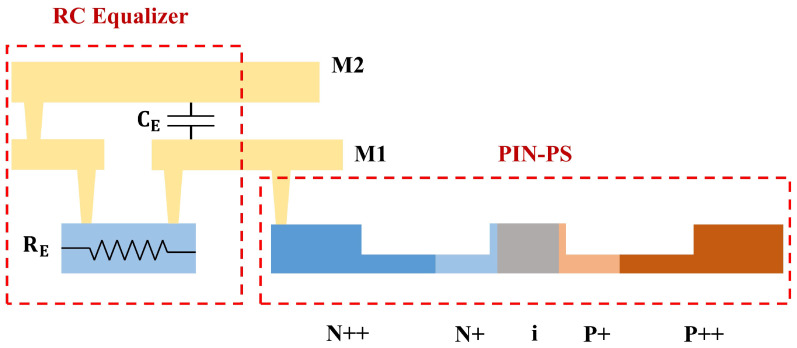
Cross-sectional structure of the PIN phase shifter integrated with passive RC equalizer.

**Figure 6 micromachines-14-01608-f006:**
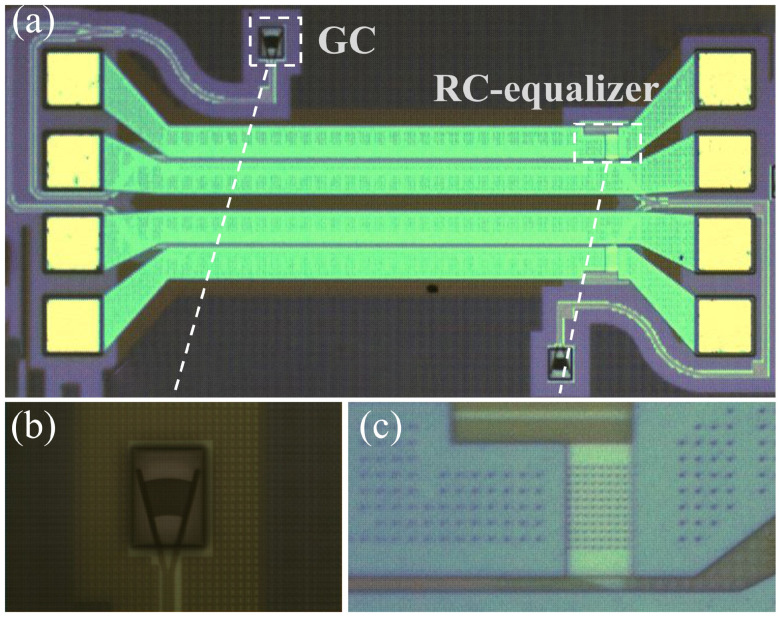
(**a**) Microscope image of the demonstrated PIN-RC modulator. (**b**) Close-up of the grating coupler (**b**) and RC equalizer (**c**).

**Figure 7 micromachines-14-01608-f007:**
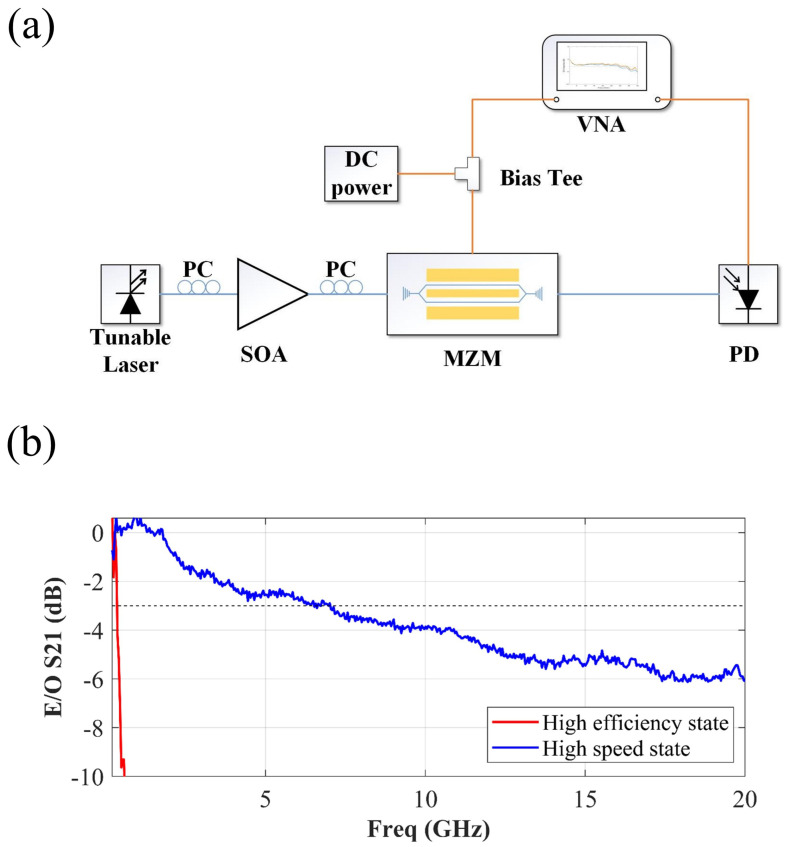
(**a**) Schematic diagrams of the experimental setup for the small-signal characterization of the PIN-RC modulator. (**b**) The measured E-O response of the PIN-RC modulator in the high-efficiency state (red line) and high-speed state (blue line).

**Figure 8 micromachines-14-01608-f008:**
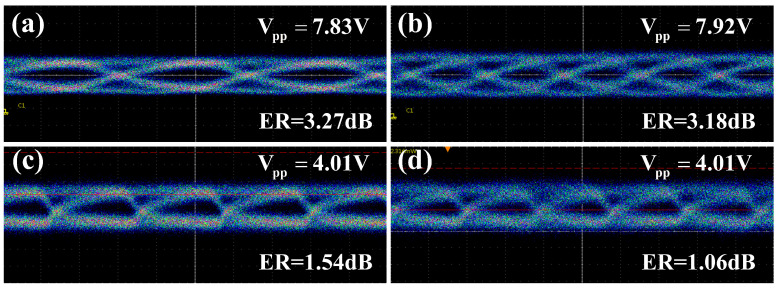
Measured eye diagrams in the high-speed state for an NRZ PRBS data modulation for different operational speeds: (**a**) 3 Gbps, (**b**) 5 Gbps, (**c**) 10 Gbps, (**d**) 15 Gbps.

**Table 1 micromachines-14-01608-t001:** Summary of the reported silicon forward-bias-based PIN Mach–Zehnder modulators.

Operating State	Reference	Structure	Insertion Loss	V${}_{\pi }$L${}_{\pi }$	Speed
	[39]	PIN + pre-emphasis signals	1.2 dB	0.29 V·cm	25 Gbps
High speed	[41]	Equalized PIN	2.78 dB	2.01 V·cm	90 Gbps
	[30]	PIN + thermal tuning	NA	0.019 V·cm	5 Gbps
High efficiency	[31]	PIN	NA	0.0025 V·cm	100 MHz
High speed and high efficiency	This work	PIN/equalized PIN	1.29 dB	1.43 V·cm, 0.0088 V·cm	15 Gbps, 200 MHz

## Data Availability

The data that support the findings of this study are available from the corresponding author upon reasonable request.

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
