# Peer review of "A Novel Silicon Forward-Biased PIN Mach–Zehnder Modulator with Two Operating States"

_micromachines, 2023, doi:10.3390/mi14081608_

Round 1

Reviewer 1 Report

The article titled " A novel silicon forward-biased PIN Mach-Zehnder modulator with two operating states “  shows results on the fabrication and characterization of a PIN modulator. The paper might be eligible for publication after addressing the following comments:

11-      Line 1: replace “demonstrated” by “demonstrate”.

22-      The authors need to justify the selection of the device doping and dimensions. The article lacks any theoretical calculations or simulations that normally precede the fabrication. This should include doping and dimensions study, which leads to the next comment.

33-      The authors need to justify their choice for choosing forward bias instead of reverse bias which is known to deliver higher speed of operation.

44-      The insertion loss is equal in two states with the value of 1.29 dB but this sounds counter intuitive unless it is measured in the OFF condition for each. But, if it is measured in the OFF state, then it is meaningless to state that it is  the same in both designs (0.0088 Vcm and 1.43 V.cm) because of course it should be the same. This needs an explanation. If it is measured in the ON state, the charge distribution is then different in each case and the insertion loss will not be the same. If my guess is correct, then what is the insertion loss for each design at the ON state.

55-      Lines 95-100:

a-        Why ReCe=RfCf? Please explain.

b-      Please justify the values given for the other parameters.

66-      The speed for the 0.0088 Vcm state is 200 MHz which sound impractical. Why do the authors feature this as a state of two for their design? This way, you can claim even much lower values of Vcm with corresponding low speeds. The 1.43 Vcm efficiency with the corresponding speed of 7 GHz sounds more of an achievement.

77-      The article lacks a table of comparison with previously published work in this field to show any advantages for the presented work.

Minor modifications are required.

Author Response

Thank you for your valuable suggestions. 

Reviewer 2 Report

The Authors demonstrate a novel silicon forward-biased PIN Mach-Zehnder Modulator. The proposed device shows promising results with large efficiency and bandwidth. As properly pointed out by the Authors, it is a Communication. Here, my comments to the manuscript:

1.       In the Introduction, MZM configuration with enhanced performance could be cited (see, i.e., High performance and tunable optical pump-rejection filter for quantum photonic systems. Optics & Laser Technology139, 106978, 2021; High extinction ratio Mach–Zehnder modulator applied to a highly stable optical signal generator. IEEE Transactions on Microwave Theory and Techniques55(9), 2007;  High extinction-ratio integrated Mach–Zehnder modulator with active Y-branch for optical SSB signal generation. IEEE Photonics Technology Letters22(12), 941-943, 2010; Low-voltage, high-extinction-ratio, Mach-Zehnder silicon optical modulator for CMOS-compatible integration. Optics express20(3), 3209-3218, 2012).

2.       The Authors should underline the approach for the device design.

3.       For PIN modulator, the study of mismatch between optical mode and RF mode is crucial to understand the bandwidth limitation. Please take into account this study.

Author Response

(The authors gave the same response as above.)

Round 2

Reviewer 1 Report

Thank you for the detailed reply.

The article is accepted in the current form.

Reviewer 2 Report

The Authors have modified the manuscript according to the comments.